# Kinetics of Glycoprotein-Specific Antibody Response in Patients with Severe Fever with Thrombocytopenia Syndrome

**DOI:** 10.3390/v14020256

**Published:** 2022-01-27

**Authors:** Hyemin Chung, Eunsil Kim, Bomin Kwon, Yeong-Geon Cho, Seongman Bae, Jiwon Jung, Min-Jae Kim, Yong-Pil Chong, Sung-Han Kim, Sang-Oh Lee, Sang-Ho Choi, Yang-Soo Kim

**Affiliations:** 1Division of Infectious Diseases, Asan Medical Center, University of Ulsan College of Medicine, Seoul 05505, Korea; newstory20@naver.com (H.C.); myboyhun@nate.com (B.K.); songman.b@gmail.com (S.B.); jiwonjung@amc.seoul.kr (J.J.); nahani99@gmail.com (M.-J.K.); drchong@amc.seoul.kr (Y.-P.C.); kimsunghanmd@hotmail.com (S.-H.K.); soleemd@amc.seoul.kr (S.-O.L.); sangho@amc.seoul.kr (S.-H.C.); 2Department of Convergence Medicine, Asan Institute for Life Sciences, Asan Medical Center, University of Ulsan College of Medicine, Seoul 05505, Korea; mykes@hanmail.net (E.K.); pooyt2002@naver.com (Y.-G.C.)

**Keywords:** SFTS, antibody, glycoprotein, cytokines, viral load

## Abstract

Severe fever with thrombocytopenia syndrome (SFTS) is an emerging tickborne disease in East Asia that is causing high mortality. The Gn glycoprotein of the SFTS virus (SFTSV) has been considered to be an essential target for virus neutralization. However, data on anti-Gn glycoprotein antibody kinetics are limited. Therefore, we investigated the kinetics of Gn-specific antibodies compared to those of nucleocapsid protein (NP)-specific antibodies. A multicenter prospective study was performed in South Korea from January 2018 to September 2021. Adult patients with SFTS were enrolled. Anti-Gn-specific IgM and IgG were measured using an enzyme-linked immunosorbent assay. A total of 111 samples from 34 patients with confirmed SFTS were analyzed. Anti-Gn-specific IgM was detected at days 5–9 and peaked at day 15–19 from symptom onset, whereas the anti-NP-specific IgM titers peaked at days 5–9. Median seroconversion times of both anti-Gn- and NP-specific IgG were 7.0 days. High anti-Gn-specific IgG titers were maintained until 35–39 months after symptom onset. Only one patient lost their anti-Gn-specific antibodies at 41 days after symptom onset. Our data suggested that the anti-Gn-specific IgM titer peaked later than anti-NP-specific IgM, and that anti-Gn-specific IgG remain for at least 3 years from symptom onset.

## 1. Introduction

Severe fever with thrombocytopenia syndrome (SFTS) is an emerging tickborne disease with mortality rates reportedly ranging from 12.3 to 32.6% [1,2,3]. SFTS is characterized by high fever, myalgia, thrombocytopenia, and leukocytopenia, and can result in multiple organ failure in severe cases [2,3]. The causative agent of SFTS is the SFTS virus (SFTSV), also known as Dabie bandavirus, of the genus *Bandavirus*, family *Phenuiviridae*, and order *Bunyavirales* [4]. The genome of this single-stranded RNA virus comprises L, M, and S segments, which encode RNA-dependent RNA polymerases, Gn and Gc glycoproteins, a nucleocapsid protein (NP) and a nonstructural protein, respectively [1].

To date, there are neither vaccinations nor specific treatments for SFTS. Although the impairment of innate and humoral immune responses and cytokine storms have been shown to play the key role in SFTS progress [5,6,7], the exact immunopathology has remained unclear. Previous studies have suggested that a failure of the virus-specific IgG response is related with poor outcomes of SFTS patients, showing the absence of NP-specific IgM and IgG antibodies and Gn-specific IgG antibodies in deceased patients [7]. Glycoproteins are essential for SFTSV to enter host cells and, therefore, constitute a pivotal target for virus neutralization [8,9]. However, previous studies on SFTS-specific virus kinetics have typically focused on NP-specific antibodies [10,11,12,13], and are limited owing to their single-center study design. Although a few studies have reported the dynamics of a specific anti-Gn antibody [7,11], to the best of our knowledge, there is a paucity of long-term follow-up studies regarding the anti-Gn-specific antibody response. Therefore, we investigated anti-Gn-specific antibody kinetics in patients with SFTS, as well as viral load and cytokine profiles, through a multicenter prospective study in South Korea. 

## 2. Materials and Methods

### 2.1. Patients and Samples

Patients aged ≥ 18 years with confirmed SFTS were enrolled at eight referral hospitals (Asan Medical Center, Ulsan University Hospital, Kyung Hee University Hospital, Hallym University Chuncheon Sacred Heart Hospital, Gangneung Asan Hospital, Gyeongsang National University Changwon Hospital, Gyeongsang National University Jinju Hospital, and Chonbuk National University Hospital) in South Korea from January 2018 to August 2021. We also enrolled patients who were diagnosed with SFTS within 3 years and recovered. Baseline characteristics, including age, sex, underlying disease, symptom onset time, and initial laboratory test results, were collected. Plasma samples were placed in ethylenediaminetetraacetic acid (EDTA)-treated collection tubes and immediately frozen at −80 °C until further analysis. Fifteen healthy volunteers had plasma samples collected as controls. This study was approved by the Institutional Review Board of all participating hospitals.

### 2.2. Quantitative RT-PCR

SFTSV RNA in plasma was extracted with a QIAmp Viral RNA Mini Kit (Qiagen, Hilden, Germany) according to the manufacturer’s instruction. cDNAs were generated via reverse transcription using the LightCycler Multiplex RNA Virus Master (Roche Diagnostics, Indianapolis, IN, USA). Viral copy numbers were determined via quantitative real-time polymerase chain reaction (qRT-PCR) with an M- and S-specific segment-based primer set as described previously [14]. Copy numbers were calculated as a ratio with respect to the standard control.

### 2.3. Enzyme-Linked Immunosorbent Assay (ELISA) Analysis of Specific Antibodies to NP and Gn

Antibody titers of IgM and IgG specific to NP and Gn were measured using ELISA. To prepare the coating antigens for ELISA, *Escherichia coli*-expressed recombinant N proteins (SFTSV KASJH) were purified, and His-tagged Gn recombinant proteins (SFTSV HN6) were purchased from eEnzyme LLC (eEnzyme LLC, Gaithersburg, MD, USA). A Nunc-Immuno Plate (Thermo Fisher Scientific, Waltham, MA, USA) was coated with a predetermined optimal quantity of antigen (100 ng per well) and incubated overnight at 4 °C. Plasma specimens were diluted to serial concentrations of 1:100, 1:200, 1:400, 1:800, and 1:1600 and analyzed in duplicate. Briefly, 100 μL of serially diluted plasma samples was incubated for 2 h at room temperature and subsequently detected using HRP-conjugated goat anti-human IgM and IgG. Absorbance was measured at 450 nm using a microplate reader (SpectraMAX190, Molecular Devices, San Jose, CA, USA). The cut-off value was set at the average optical density (OD) value of healthy volunteers plus three times the standard deviation (SD; mean + 3 × SD). The sample was considered positive if an OD_450_ value above the cut-off value was yielded.

### 2.4. Cytokine Measurements

Inflammatory mediators were measured in the plasma samples of patients and controls using a ProcartaPlex Multiplex Immunoassay (Thermo Fisher Scientific, Waltham, MA, USA) and a Luminex MAGPIX^®^ System (Merck Millipore, Burlington, MA, USA) according to the manufacturers’ instructions. The following inflammatory mediators were measured: interferon (IFN)-α, IFN-γ, interleukin (IL)-6, IL-8, IL-10, IL-17A, IFN-γ-induced protein (IP)-10, and monocyte chemotactic protein (MCP). 

### 2.5. Statistical Analysis

Statistical analysis was performed using the Statistical Package for the Social Sciences (SPSS version 23.0; IBM Corp., Armonk, NY, USA). Categorical variables were compared using the chi-square test or Fisher’s exact test as appropriate, and continuous variables were compared using Student’s *t*-test and the Mann–Whitney U test as appropriate. Spearman’s test was used to calculate the correlation coefficient between viral RNA load/cytokine/chemokine levels and antibody titer level. *p*-values of <0.05 were considered to be statistically significant.

## 3. Results

### 3.1. Baseline Characteristics of Patients

A total of 34 patients (30 survivors and 4 non-survivors) with confirmed SFTS were enrolled in this study, and 111 plasma samples were collected. Of the 34 patients, 13 recovered patients were enrolled, and plasma samples were obtained after 30 days from symptom onset. The median follow-up time was 370.5 days (interquartile range (IQR), 28.3–766.3) after symptom onset. Overall, the median age of patients was 66.5 (IQR, 58.0–73.3), and 15 of the 34 patients (44.1%) were male. Fifteen patients (55.9%) had underlying diseases; the most common underlying diseases were diabetes mellitus (35.3%) and hypertension (32.4%). Insect bite sites were observed in eight patients (23.5%). Median initial white blood cell counts (×10^3^/µL) and platelet counts (×10^3^/µL) were 2.3 (IQR, 1.1–5.8) and 59.0 (47.5–92.5), respectively. There were no significant differences in baseline characteristics between survivors and non-survivors. 

### 3.2. Anti-Gn and NP Specific IgM and IgG 

Figure 1 shows the kinetics of anti-Gn-specific IgM and IgG and anti-NP-specific IgM and IgG. Both anti-Gn-specific IgM and anti-NP-specific IgM were detected at days 5–9 from symptom onset. The titers of anti-Gn-specific IgM and anti-NP-specific IgM peaked at days 15–19 and days 5–9 from symptom onset, respectively. Anti-Gn-specific IgM and anti-NP-specific IgM became undetectable after 1 months from symptom onset in all patients. Both anti-Gn- and NP-specific IgG could be detected from days 1 to 4 following symptom onset. The median time of anti-Gn IgG and anti-NP-specific IgG seroconversion was at 7.0 days, with an IQR of 5.8–15.3, and 7.0 days, with an IQR of 5.3–9.0, respectively. The anti-NP-specific IgG titer tended to be increased until 5–9 months after symptom onset and then decreased, but remained above cut-off levels at 35–39 months after symptom onset. Contrastingly, the anti-Gn-specific IgG titer remained high at 35–39 months. During the follow-up period, only one patient lost anti-Gn-specific antibodies, occurring at 41 days after symptom onset. 

Table 1 shows the seropositive rates of anti-Gn-specific IgM and IgG, and anti-NP-specific IgM and IgG. The seropositive rates of anti-Gn-specific IgM and anti-NP-specific IgM were the highest on days 15–19, and days 10–14, respectively. Anti-Gn-specific IgG and anti-NP-specific IgG seropositive rates became 100% on days 15–19 and days 10–14, respectively. There were no significant differences in the seropositive rates of anti-Gn-specific IgM (survivors, 1/13 (7.7%) vs. non-survivors, 1/3 (33.3%), *p* = 0.35), anti-Gn-specific IgG (6/13 (46.2%) vs. 2/3 (66.7%), *p* > 0.99), anti-NP-specific IgM (5/13 (38.5%) vs. 2/3 (66.7%), *p* = 0.55), and anti-NP-specific IgG (10/13 (76.9%) vs. 2/3 (66.7%), *p* > 0.99) between surviving and deceased patients within 1 week from symptom onset. Among the four deceased patients, one, two, two, and three patients had positive anti-Gn-specific IgM, IgG, anti-NP-specific IgM, and IgG, respectively, within 2 weeks after symptom onset. 

### 3.3. Viral Load

Viral loads were highest on days 1–4 from symptom onset, and decreased gradually throughout the course of hospitalization (Figure 2). The mean viral load in deceased patients at days 5–9 from symptom onset was significantly higher than that in survivors (M segment; survivors, 3.07 ± 0.75 vs. non-survivors, 5.43 ± 1.16, *p* = 0.002; S segment, 3.00 ± 0.69 vs. 5.49 ± 0.66, *p* = 0.001). In survivors, the OD_450_ values of the anti-Gn-specific IgG titer (r = −0.302, *p* = 0.03) and that of the anti-NP-specific IgG titer (r = −0.407, *p* = 0.003) at the dilution level of 1:100 in ELISA correlated negatively with the level of serum SFTSV RNA in the correlation analysis.

### 3.4. Cytokines and Chemokines

A total of 74 plasma samples from 17 patients and 15 plasma samples from 15 healthy volunteers were measured for cytokine levels. To compare cytokine and chemokine levels between patients with SFTS and healthy volunteers, cytokine and chemokine levels obtained from the first drawn plasma samples during admission in patients with SFTS were compared with that of healthy volunteers. Mean levels of IFN-α (patients, 235.60 pg/mL vs. healthy volunteers, 0.12 pg/mL, *p* = 0.01), IFN-γ (28.91 pg/mL vs. 1.58 pg/mL, *p* < 0.001), IL-10 (83.02 pg/mL vs. 1.09 pg/mL, *p* < 0.001), and IL-6 (99.8 pg/mL vs. 6.67 pg/mL, *p* = 0.01) were significantly higher in SFTS patients than those in healthy volunteers. 

IFN-α (non-survivor, 1387.2 pg/mL vs. survivor, 154.8 pg/mL, *p* = 0.03) and IFN-γ (162.7 pg/mL vs. 15.1 pg/mL, *p* = 0.03) levels in the deceased patients within one week (days 1–7) after symptom onset were significantly higher than those in survivors (Figure 3). In survivors, the levels of IFN-α, IFN-γ, IL-10, IL-6, IL-8, IP-10, and MCP-1 were mildly elevated during the early phase of the SFTS clinical course, and then decreased throughout the remainder of the course of the disease. IL-17A levels did not show significant changes during the clinical course of the disease. In deceased patients, IL-10, IL-6, IL-8, IP-10, and MCP-1 levels were markedly increased at 2 weeks (days 8–14) after symptom onset, whereas IFN-α and IFN-γ levels decreased at 2 weeks. 

In survivors, the anti-NP-specific IgM titer correlated positively with IL-6 (r = 0.348, *p* = 0.01), IP-10 (r = 0.608, *p* < 0.001), whereas the anti-NP-specific IgM titer correlated with IFN-γ (r = 0.297, *p* = 0.04), IL-8 (r = 0.286, *p* = 0.04), and MCP-1 (r = 0.282, *p* = 0.048) levels. In non-survivors, the anti-Gn-specific IgM titer correlated positively with IFN-α (r = 0.920, *p* = 0.03), and IFN-γ (r = 0.908, *p* = 0.03), whereas the anti-Gn-specific IgG titer negatively correlated with IL-6 (r = −0.931, *p* = 0.02), IL-8 (r = −0.893, *p* = 0.04), IP-10 (r = −0.956, *p* = 0.01), and MCP-1 (r = −0.899, *p* = 0.04).

## 4. Discussion

In this study, we evaluated the kinetics of anti-Gn-specific antibodies in SFTS patients throughout a three-year follow-up period. Anti-Gn-specific IgM antibodies were detectable later than NP-specific IgM, and the titer of anti-Gn-specific IgM peaked at days 15–19 after symptom onset. Anti-Gn-specific IgG antibodies were maintained for three years after symptom onset with a high titer. Additionally, the viral load of SFTSV was significantly higher in deceased patients than in survivors, and the IFN-α and IFN-γ levels of the deceased patients at one week after symptom onset were significantly higher than that of survivors. 

Song et al. reported the dynamics of anti-Gn-specific IgM and IgG from 3 to 18 days after symptom onset [7]. They found that patients tested positive for an anti-Gn-specific IgM in the first week, with the anti-Gn-specific IgM becoming undetectable in the second week. Additionally, the anti-Gn-specific IgG titers were increased during the clinical course. However, our study revealed a delayed seroconversion of anti-Gn-specific IgM, with the peak titer at days 15–19 after symptom onset. This discrepancy in the detection time of anti-Gn-specific IgM seroconversion could be explained by the different experimental methods of ELISA, such as the dilution factor. The differences in baseline characteristics, including age, male, and underlying diseases, and the severity of SFTS also could be reasons for the discrepancy. In addition, both studies included a relatively small number of cases, necessitating further large-scale studies with a highly sensitive IgM-capture ELISA test to understand the kinetics of anti-Gn-specific antibodies. Although neutralizing antibodies to SFTSV and anti-NP-specific IgG were reportedly maintained for up to four years after hospitalization and eight years after illness [13,15], there have been no reports of a long-term follow-up response in anti-Gn-specific antibodies to the best of our knowledge. The glycoprotein of SFTSV is responsible for receptor binding and membrane fusion, and is considered to be a target for neutralization [16]. Kim et al. revealed that anti-Gn antibodies from convalescent SFTS patients had therapeutic efficacy in a SFTS-infected mouse model [17]. Song et al. demonstrated the neutralization activity of Gn-specific antibodies from convalescent patients [7]. In addition, a recombinant viral vector-based vaccine expressing the SFTSV glycoprotein or DNA vaccine encoding the SFTSV glycoprotein has been reported as a vaccine candidate for SFTS [18]. If further evidence of a protective effect of anti-Gn-specific IgG is provided by the neutralizing antibody assay, our Gn-specific antibody response data might be helpful in choosing a potential donor of convalescent plasma therapy or developing a vaccination strategy. 

Failure in B-cell class switching with the absence of virus-specific antibody are suggested to be key factors for determining the disease severity and clinical outcomes of SFTS [7,11,19]. Song et al. revealed that anti-NP-specific IgM and IgG and anti-Gn-specific IgG were negative in deceased patients with SFTS, and that the intracellular staining of IgG and IgM in plasmablasts were negative in fatal SFTS patients [7]. A recent single-cell transcriptome sequencing study demonstrated that the lineage of B cells includes SFTSV targets [19]. However, among the four deceased patients in our study, anti-Gn-specific IgM and IgG were positive in 25.0% and 50.0% patients, respectively, and anti-NP-specific IgM and IgG were positive in 50.0% and 75.0% patients, respectively. Because the exact pathogenesis of SFTS is not clear and is likely complex, factors other than B cell immunity function, including the underlying characteristics of patients [3], cytokine storms [20,21], high viral loads [20,22], the impairment of innate immune response [6], or alterations in T cell response [7], may have contributed to the fatal outcomes observed in our patients. 

The limitation of this study is that we did not conduct the neutralization antibody assay. As there were no significant differences in the seropositive rates of anti-Gn-specific antibodies between survivors and non-survivors, further study including the neutralizing antibody assay are needed to know the possibility of a protective effect of anti-Gn-specific IgG against SFTS reinfection. Second, OD_450_ values in the early phase of SFTS were relatively high, possibly due to the non-specific reaction. Further study including B cell ELISPOT for the detection of Gn-specific antibody secreting cells are needed.

In conclusion, we found that anti-Gn-specific IgM is detected later than anti-NP-specific IgM, and patients remain positive for anti-Gn-specific IgG for up to three years following the onset of their symptoms. An anti-NP-specific IgM ELISA rather than an anti-Gn-specific IgM ELISA could be more helpful for SFTS diagnosis in the early phase. Our study might aid in the development of future therapeutics or vaccinations for SFTS, if further evidence of a protective effect of anti-Gn-specific IgG for SFTS reinfection is provided.

## Figures and Tables

**Figure 1 viruses-14-00256-f001:**
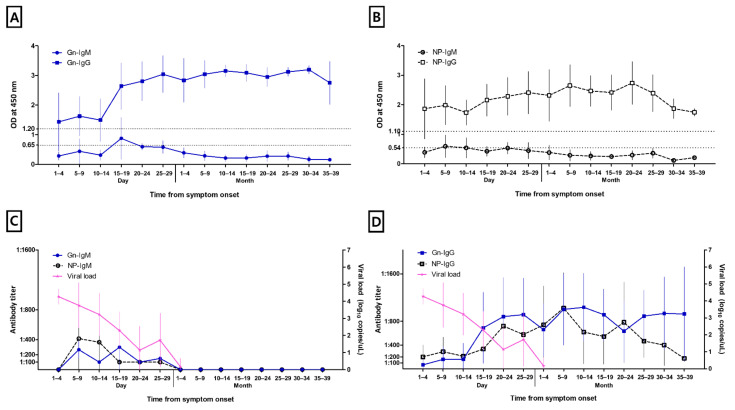
Kinetics of severe fever with thrombocytopenia syndrome (SFTS)-specific antibodies measured by enzyme-linked immunosorbent assay: (**A**) anti-Gn glycoprotein-specific antibody IgM and IgG; (**B**) anti-nucleocapsid protein (NP)-specific antibody IgM and IgG; (**C**) anti-Gn glycoprotein-specific antibody IgM, anti-NP-specific IgM, and SFTSV load; and (**D**) anti-Gn glycoprotein-specific antibody IgG, anti-NP-specific IgG, and SFTSV load.

**Figure 2 viruses-14-00256-f002:**
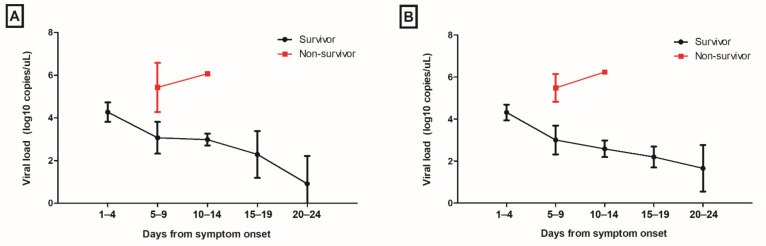
Kinetics of viremia in patients with severe fever with thrombocytopenia syndrome (SFTS): (**A**) M segment and (**B**) S segment.

**Figure 3 viruses-14-00256-f003:**
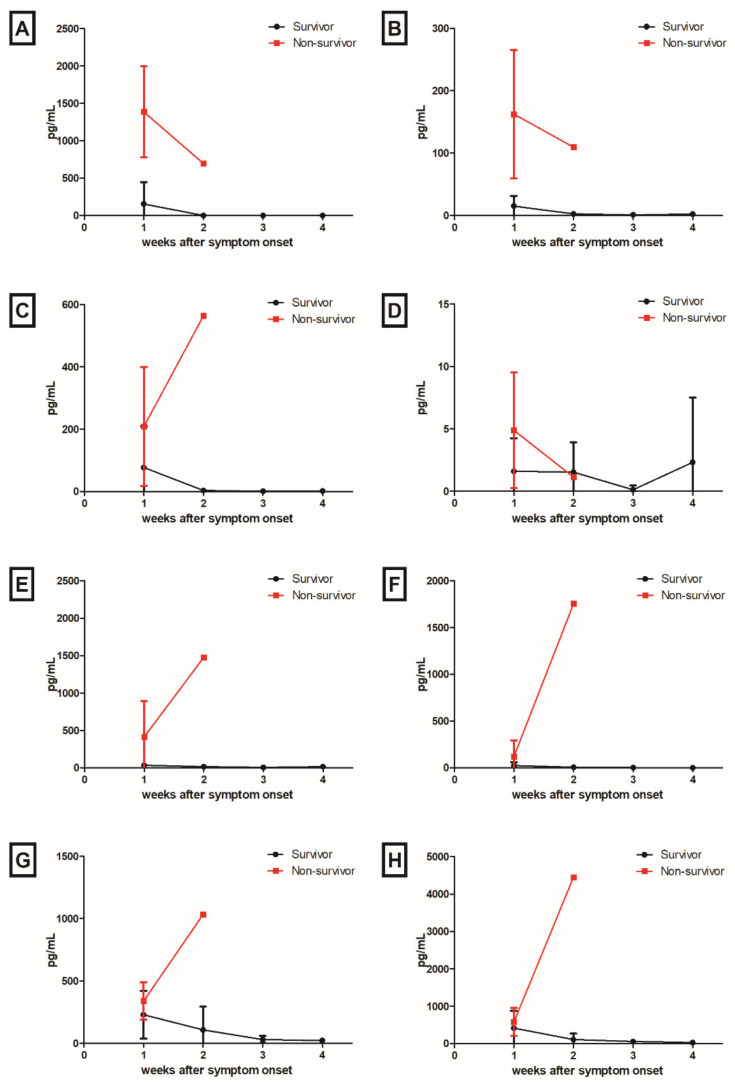
Kinetics of cytokines and chemokines in patients with SFTS: (**A**) interferon (IFN)-α; (**B**) IFN-γ; (**C**) interleukin (IL)-10; (**D**) IL-17A; (**E**) IL-6; (**F**) IL-8; (**G**) IFN-γ-induced protein (IP)-10; and (**H**) monocyte chemotactic protein (MCP)-1 levels.

**Table 1 viruses-14-00256-t001:** Seropositive rates of patients with severe fever with thrombocytopenia syndrome *.

Days from Symptom Onset	Gn-IgM	Gn-IgG	NP-IgM	NP-IgG
Days 1–4	0/3 (0)	1/3 (33.3)	0/3 (0)	2/3 (66.7)
Days 5–9	3/16 (18.8)	9/16 (56.3)	6/16 (37.5)	14/16 (87.5)
Days 10–14	1/9 (11.1)	6/9 (66.7)	4/9 (44.4)	9/9 (100.0)
Days 15–19	4/9 (44.4)	9/9 (100.0)	2/9 (22.2)	9/9 (100.0)
Days 20–24	2/5 (40.0)	5/5 (100.0)	2/5 (40.0)	5/5 (100.0)
Days 25–29	2/8 (25.0)	8/8 (100.0)	2/8 (25.0)	8/8 (100.0)

abbreviations: Gn-IgM, anti-Gn glycoprotein immunoglobulin M; IgG, immunoglobulin G; NP, nucleocapsid protein. * Numerator and denominator represent the number of seropositivity cases and the total number of available data points, respectively.

## Data Availability

The data presented in this study are available in this article.

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
