# Peer review of "Kinetics of Glycoprotein-Specific Antibody Response in Patients with Severe Fever with Thrombocytopenia Syndrome"

_viruses, 2022, doi:10.3390/v14020256_

Round 1

Reviewer 1 Report

The authors studied the sequential profiles on SFTSV antibodies IgG and IgM to Gn and NP of SFTSV using the ELISA system. The authors studied the viral loads of SFTSV in patients with SFTS in an acute phase of the disease. Furthermore, the authors measured the cytokine and chemokine profiles. The authors compared the level of viral loads and cytokine/chemokine levels between patients with SFTS who died and those who did not, indicating that these indices in fatal patients were significantly higher than those in patients who recovered.

The major conclusion might be that anti-Gn specific IgM became detectable within 5-9 days from the disease onset and peaked at day 15-19 from the onset. Anti-NP specific IgM was detected at days 5-9. Median seroconversion times of both anti-Gn and NP specific IgG were 7.0 days. High anti-Gn specific IgG titers were maintained until 35-39 months after the disease onset. The main conclusion written in this manuscript might be that “anti-Gn specific IgM titer peaked later that anti-NP specific IgM, and that anti-Gn specific IgG remain for 3 years from symptom onset.

Major comments

  1. It is written that this study has been conducted in a prospective manner. However, there are the participants, from whom plasma samples were collected only in a convalescent phase. This reviewer considers that this event should not happen, if it had been designed to collect plasma samples not only in a convalescent phase but also in an acute phase, when the participants were hospitalized. Explanation is necessary.
  2. The assay for detection of IgM antibodies against Gn and NP were based on the direct ELISA, in which antigens were coated on the bottom of ELISA and secondary antibodies against human anti-IgM and human anti-IgG antibodies were used, respectively. Both the IgM and IgG both to Gn and IgG to NP had become detectable even in a very acute phase (days-1-4). The OD values at this point are over 1.00 (Figure 1A and 1B), although the dilution level is not mentioned. This reviewer considers that the value is very high possibly due to the non-specific reaction. The OD values in each positive antigen-coated well should be subtracted by those in negative-antigen-coated well. This reviewer considers that IgM ELISA is better be based on the IgM-capture ELISA to make the IgM assay system more specific and reliable.
  3. This reviewer considers that neutralization antibody assay results should be included.
  4. This reviewer does not understand the reason why the SFTSV loads and chemokine/cytokine levels in comparison between fatal cases and non-fatal cases are included in this study. Are there any association in the antibody response and viral loads/cytokine/chemokine responses between fatal cases and non-fatal case? Explanation is necessary.
  5. This reviewer considers that the study design is not appropriate for drawing conclusions on Ig G and IgM antibody response to Gn and NP of SFTSV in patients with SFTS so much.

Specific comments

  1. The title: The description “response to severe fever with thrombocytopenia syndrome” does not make sense. “response in patients with severe fever with thrombocytopenia syndrome” might be better.
  2. Line 27: “remain for 3 years from symptom onset” might be “remain at least for 3 years from symptom onset”.
  3. Lie 38: The reference [1] does not seem appropriate for this sentence.
  4. Line 39: This reviewer prefers to use “specific treatments for SFTS” rather to “effective treatments for SFTS”.
  5. Line 40: “poorly characterized” might be to much saying.
  6. Line 54: “Adult” should be deleted because the age of the participants is defined.
  7. Line 63: Please make it clear whether the ethical institutional approval had been obtained only from Asan Medial University or obtained from all the ethical research review boards of all the institutes participated. Was informed consent obtained from all the participants?
  8. Line 68 “quantified” might be “quantitative”.
  9. Line 73: “antibodies for NP ---” might be “antibodies to NP ---”.
  10. Line 91: “LuminexTM200TM” is not a manufacturer’s instruction. Correction is needed.
  11. Line 104: It is written that plasma samples were obtained only during the convalescent phase following discharge. If this study had been conducted in a prospective manner, this reviewer considers that this kind of event should not happen. Explanation is necessary.
  12. Line 120: The word “disappeared” is not appropriate for this description. “became undetectable” is better.
  13. Line 123-124: The description “Anti-NP specific IgG titer was increased until 5-9 month after symptom onset and then decreased” might be misunderstanding. The titers at each time point are different with statistically significant? Figure 1D does not indicate the result.
  14. Line 128-133: The calculation methods for antibody positive rates for each antigen at each time point should be mentioned. The number of samples tested at each time points should be written.
  15. Line 134-135: The data for the description by this sentence should be included. According to the previous reports, antibody responses were very weak in fatal patients.
  16. Line 166: “higher” should be “significantly higher”?
  17. Line 199: “antibodies for SFTS” should be “antibodies to SFTSV”.
  18. Line 202: “glycoprotein of SFTS” should be “glycoprotein of SFTSV”.
  19. References: The form of reference should be that of the journal, to which the manuscript is submitted.

Reviewer 2 Report

  1. Review.

Journal: Viruses

Manuscript ID; Viruses-1499650-rev1

Title; Kinetics of glycoprotein-specific antibody response to severe fever with thrombocytopenia syndrome

Authors; Hyemin Chung et al.,

In this manuscript, the authors investigated the dynamics of anti-Gn and anti-NP specific antibodies in SFTS patients follow-up for three-years. Furthermore, they compared the viral load and cytokine/chemokines dynamics between survivors and non-survivors.

They confirmed that patients remain positive for anti-Gn specific IgG for up to three years following the onset of their symptoms. Contrary to past report of Song et al., anti-Gn anti-NP specific antibodies were positive in 50-75% of deceased patients with SFTS.

This manuscript is well prepared and revealed the long lasting of SFTS antibodies over three-years. Therefore, it can be accepted to be published in Viruses with some changes.

Major points:

  1. The authors suggested the possibility of a protective effect against SFTS reinfection, because anti-Gn specific IgG antibody was maintained three years after symptom onset with a high titer in Discussion section. However, there were no significant differences in seropositive rates or titers of anti-Gn antibodies between survivors and non-survivors. To get rid of these discrepancies, the neutralizing tests should be conducted. The authors should delete this suggestion or try to measure neutralizing antibody for SFTSV.

  1. At Line 223-231, the authors explained the weak points of this survey. If the authors explain the limitation of this research, the explanation of no-NT assay Should be added in this explain. Certainly these points were weakness of this manuscript, however these explanations were just excuses for reviewers. I ask authors to remove theses excuses.

  1. To clarify the kinetics of antibody and viral loads of SFTS patients, SFTS-antibody kinetics should be overlayed in Figure 2.

  1. The data for seropositive rate of SFTS patients described at Line127-133 should be Table or Figure for a better understanding of readers.

Minor points:

  1. Describe the origin of antigens( what viral strain did you use)at Line76-77.
  2. Leave space between Line: Line88, Line95.

Author Response

Reviewer: 2

Comments to the Author

In this manuscript, the authors investigated the dynamics of anti-Gn and anti-NP specific antibodies in SFTS patients follow-up for three-years. Furthermore, they compared the viral load and cytokine/chemokines dynamics between survivors and non-survivors.

They confirmed that patients remain positive for anti-Gn specific IgG for up to three years following the onset of their symptoms. Contrary to past report of Song et al., anti-Gn anti-NP specific antibodies were positive in 50-75% of deceased patients with SFTS.

This manuscript is well prepared and revealed the long lasting of SFTS antibodies over three-years. Therefore, it can be accepted to be published in Viruses with some changes.

Major points:

  1. The authors suggested the possibility of a protective effect against SFTS reinfection, because anti-Gn specific IgG antibody was maintained three years after symptom onset with a high titer in Discussion section. However, there were no significant differences in seropositive rates or titers of anti-Gn antibodies between survivors and non-survivors. To get rid of these discrepancies, the neutralizing tests should be conducted. The authors should delete this suggestion or try to measure neutralizing antibody for SFTSV.

à Thank you for reviewing our study in detail and providing helpful comments. Because of limitation of our laboratory resources, we could not conduct the neutralizing antibody assay. As the reviewer suggested, we deleted the suggestion of the possibility of a protective effect against SFTS reinfection. We revised the manuscript as follows:

[Discussion, page 6, line 215–216]

“Anti-Gn specific IgG antibody was maintained three years after symptom onset with a high titer.”  

[Discussion, page 7, line 254–258]

“The limitation of this study is that we did not conduct the neutralization antibody assay. As there were no significant differences in seropositive rates of anti-Gn antibody between survivors and non-survivors, further study including the neutralizing antibody assay are needed to know the possibility of a protective effect of anti-Gn specific IgG against SFTS reinfection.” 

  1. At Line 223-231, the authors explained the weak points of this survey. If the authors explain the limitation of this research, the explanation of no-NT assay Should be added in this explain. Certainly these points were weakness of this manuscript, however these explanations were just excuses for reviewers. I ask authors to remove theses excuses.

à As the reviewer suggested, we removed limitation points what we described in the previous manuscript and added the explanation of no-NT assay in the limitation session as follows:

[Discussion, page 7, line 254–258]

“The limitation of this study is that we did not conduct the neutralization antibody assay in this study. As there were no significant differences in seropositive rates of anti-Gn antibody between survivors and non-survivors, further study including the neutralizing antibody assay are needed to know the possibility of a protective effect of anti-Gn specific IgG against SFTS reinfection.”  

  1. To clarify the kinetics of antibody and viral loads of SFTS patients, SFTS-antibody kinetics should be overlayed in Figure 2.

à As the reviewer’s suggestion, we revised the figure as follows:

[Figure, page 4, line 156–160]

Figure 1. Kinetics of severe fever with thrombocytopenia syndrome (SFTS)-specific antibodies measured by enzyme-linked im-munosorbent assay: (A) anti-Gn glycoprotein-specific antibody IgM and IgG; (B) anti-nucleocapsid protein (NP) specific antibody IgM and IgG; (C) anti-Gn glycoprotein-specific antibody IgM, anti-NP specific IgM, and SFTSV load; and (D) anti-Gn glycoprotein-specific antibody IgG, anti-NP specific IgG, and SFTSV load.

  1. The data for seropositive rate of SFTS patients described at Line127-133 should be Table or Figure for a better understanding of readers.

à As the reviewer’s suggestion, we add a new table revealing seropositive rate of patients. We revised the manuscript as follows:

[Results, page 4, line 140–146]

“Seropositive rates of anti-Gn specific IgM and IgG at days 1–4, 5–9, 10–14, 15–19, and 25–29 from symptom onset were 0% (0/3), 18.8% (3/16), 11.1% (1/9), 44.4% (4/9), 40.0% (2/5), and 25.0% (2/8) and 33.3% (1/3), 56.3% (9/16), 66.7% (6/9), 100.0% (9/9), 100.0% (5/5), and 100.0% (8/8), respectively. Positive rates of anti-NP specific IgM and IgG at days 1–4, 5–9, 10–14, 15–19, and 25–29 from symptom onset were 0% (0/3), 37.5% (6/16), 44.4% (4/9), 22.2% (2/9), 40.0% (2/5), and 25.0% (2/8) and 66.7% (2/3), 87.5% (14/16), 100.0% (9/9), 100.0% (9/9), 100.0% (5/5), and 100.0% (8/8), respectively (Table 1).”

[Table, page 4–5, line 162–165]

Table 1. Seropositive rates of patients with severe fever with thrombocytopenia syndrome.*

Days from symptom onset

Gn-IgM

Gn-IgG

NP-IgM

NP-IgG

Day 1-4

0/3 (0)

1/3 (33.3)

0/3 (0)

2/3 (66.7)

Day 5-9

3/16 (18.8)

9/16 (56.3)

6/16 (37.5)

14/16 (87.5)

Day 10-14

1/9 (11.1)

6/9 (66.7)

4/9 (44.4)

9/9 (100.0)

Day 15-19

4/9 (44.4)

9/9 (100.0)

2/9 (22.2)

9/9 (100.0)

Day 20-24

2/5 (40.0)

5/5 (100.0)

2/5 (40.0)

5/5 (100.0)

Day 25-29

2/8 (25.0)

8/8 (100.0)

2/8 (25.0)

8/8 (100.0)

Abbreviations: Gn-IgM, anti-Gn glycoprotein immunoglobulin M; IgG, immunoglobulin G; NP, nucleocapsid protein. *Numerator and denominator represent a number of seropositivity and a number of available data, respectively.

Minor points:

  1. Describe the origin of antigens( what viral strain did you use)at Line76-77.

à SFTS virus KASJH was used as the origin of N antigen and SFTS virus HN6 was used as the origin of Gn antigen. We revised the manuscript as follows:

[Methods, page 2, line 81–83]

“To prepare coating antigens for ELISA, Escherichia coli-expressed recombinant N protein (SFTS virus KASJH) were purified, and His-tagged Gn recombinant proteins (SFTS virus HN6) were purchased from eEnzyme LLC (Gaithersburg, USA).”

  1. Leave space between Line: Line88, Line95.

à Thank you again for reviewing our study and providing critically helpful comments. As the reviewer suggested, we leaved space in the revised manuscript.

Round 2

Reviewer 1 Report

The authors have made revisions. The authors studied the sequential profiles on SFTSV antibodies IgG and IgM to Gn and NP of SFTSV using the ELISA system. The authors studied the viral loads of SFTSV in patients with SFTS in an acute phase of the disease. Furthermore, the authors measured the cytokine and chemokine profiles. The authors compared the level of viral loads and cytokine/chemokine levels between patients with SFTS who died and those who did not, indicating that these indices in fatal patients were significantly higher than those in patients who recovered.

The major conclusion might be that anti-Gn specific IgM became detectable within 5-9 days from the disease onset and peaked at day 15-19 from the onset. Anti-NP specific IgM was detected at days 5-9. Median seroconversion times of both anti-Gn and NP specific IgG were 7.0 days. High anti-Gn specific IgG titers were maintained until 35-39 months after the disease onset. The main conclusion written in this manuscript might be that “anti-Gn specific IgM titer peaked later that anti-NP specific IgM, and that anti-Gn specific IgG remain for 3 years from symptom onset.

Major comments

  1. To this reviewer, the main conclusion that “anti-Gn specific IgM titer peaked later that anti-NP specific IgM” seems not so worthy of reporting. This reviewer does not consider that this conclusion is of value for publication. If it is of worth for publication, the authors should explain why this conclusion is important for understanding the nature and characteristics of SFTS clearly.
  2. This reviewer recommends again that the antibodies to SFTSV in patients with SFTS in the early phase of the disease with 4 days from disease onset should be tested with neutralization antibody assay or indirect immunofluorescence assay, if possible, to make this manuscript more attractive.

Specific comments

  1. Line 40: The sentence structure of “no --- nor” should be “neither --- nor”.
  2. Line 84: “(Gaithersburg, USA) should be “(company name, city of location, state in the USA)”.
  3. Line 99: “Massachusetts” should be abbreviated.
  4. Line 125: “(µL) might be “(10 sup3sup/µL)”.
  5. Line 143-149: This phrase is a duplicate description for Table1. Furthermore, the sentence is too complex to understand. Modification is necessary.
  6. Line 150-154: In this phrase, the difference in the antibody positive rate of Gn-IgG, Gn-IgM, NP-IgG, and NP-IgM in patients with SFTS within 7 days from the disease onset between survivors and non-survivors. This reviewer considers that the number of patients of non-survivors was too small to draw this conclusion. The difference should be statistically assessed.
  7. Line 175: “higher in survivors” is better to be corrected to “higher than that in survivors”.
  8. Line 177: “OD450 values of anti-Gn specific IgG titer” does not make sense. It should be “OD 450 values of anti-Gn specific IgG at the dilution level of *** in ELISA”.
  9. Line 201: “Similar” to what?
  10. Line 231-233: The discussion may be difficult to be understood by the readers including this reviewer. What does the term “different experimental methods” stand for? Please explain how the patient characteristics different from those of the previous reports.
  11. Line 236: “SFTS” should be “SFTSV”.
  12. Line 243-245: It is written that the data on the antibody response in this study aid in the development of future therapeutics or vaccinations for SFTS. The explanation on how the data aid in the development of future therapeutics and vaccinations.
  13. Line 255: “not unclear” indicates “clear”?
  14. Line 275-277: Conclusion should be written more carefully to include why this study is important.
  15. References: The form of references exactly be that of the journal, to which this manuscript is submitted. “The New England journal of medicine” should be “N Engl J Med”, for instance.

Round 3

Reviewer 1 Report

The quality of this manuscript has been improved by the revision process.